# Repeatability and timing of tropical influenza epidemics

**Joseph L. Servadio** [1] *, **Pham Quang Thai**[2,3], **Marc Choisy**[4,5], **Maciej F. Boni**[1,5]

**1** Center for Infectious Disease Dynamics and Department of Biology, Pennsylvania State University, University Park, Pennsylvania, United States of America, **2** National Institute of Hygiene and Epidemiology, Hanoi, Vietnam, **3** School of Preventative Medicine and Public Health, Hanoi Medical University, Hanoi, Vietnam, **4** Oxford University Clinical Research Unit, Ho Chi Minh City, Vietnam, **5** Centre for Tropical Medicine and Global Health, Nuffield Department of Medicine, University of Oxford, Oxford, United Kingdom

* jjs7684@psu.edu

**Data Availability Statement:** The authors confirm that all data underlying the findings are fully available without restriction. All relevant data and code from this study are available at github.com/ jlservadio/Influenza_Seas_VN.

## Abstract

Much of the world experiences influenza in yearly recurring seasons, particularly in temperate areas. These patterns can be considered repeatable if they occur predictably and consistently at the same time of year. In tropical areas, including southeast Asia, timing of influenza epidemics is less consistent, leading to a lack of consensus regarding whether influenza is repeatable. This study aimed to assess repeatability of influenza in Vietnam, with repeatability defined as seasonality that occurs at a consistent time of year with low variation. We developed a mathematical model incorporating parameters to represent periods of increased transmission and then fitted the model to data collected from sentinel hospitals throughout Vietnam as well as four temperate locations. We fitted the model for individual (sub)types of influenza as well as all combined influenza throughout northern, central, and southern Vietnam. Repeatability was evaluated through the variance of the timings of peak transmission. Model fits from Vietnam show high variance (sd = 64–179 days) in peak transmission timing, with peaks occurring at irregular intervals and throughout different times of year. Fits from temperate locations showed regular, annual epidemics in winter months, with low variance in peak timings (sd = 32–57 days). This suggests that influenza patterns are not repeatable or seasonal in Vietnam. Influenza prevention in Vietnam therefore cannot rely on anticipation of regularly occurring outbreaks.

## Author summary

Much of the world experiences influenza in yearly recurring seasons, particularly in temperate locations. Such seasons occur each year with predictable timing. In tropical locations, even if there are influenza epidemics each year that are labeled influenza seasons, their timing is less consistent and predictable. Here, we define repeatability to refer to regular epidemics that are consistent and predictable in their timing, such as a yearly epidemic occurring during the same time of year. We measured the repeatability of influenza in Vietnam using data collected from ten years of sentinel surveillance. Using a mathematical model, we measured repeatability by estimating the timings of epidemics and how

**Funding:** This study was funded by the National Institutes of Health grant F32AI167600 (JLS) and the National Institutes of Health Centers of Excellence in Influenza Research and Surveillance contract HHS N272201400007C (MFB). The funders had no role in the study design, data collection and analysis, decision to publish, or preparation of the manuscript.

**Competing interests:** The authors declare that they have no competing interests.

they spread throughout the year. We found that influenza epidemics in northern, central, and southern Vietnam varied widely in their timing; even if an epidemic occurred each year, their timings were typically inconsistent. By contrast, influenza epidemics were highly repeatable in four temperate locations examined. Through this, we show that influenza in Vietnam does not show consistent timings, making preparedness efforts such as vaccination campaigns difficult to design.

## 1. Introduction

Influenza is consistently regarded as a major public health concern globally, causing high burden despite efforts to update and distribute vaccines prior to upcoming influenza seasons [1]. It has been estimated that, annually, between 10 and 20 percent of the global population is infected with influenza [2], and between 250,000 and 650,000 deaths [3–5] occur. Morbidity and mortality differ across age groups, with young children and elderly adults experiencing highest burden [6,7]. In addition to morbidity and mortality, influenza has been shown to adversely impact the global economy [8]. The public health importance of influenza is apparent both during typical influenza seasons as well as during major pandemics, as seen notably in 1918 as well as in 1957, 1968, and most recently in 2009 [9].

In much of the world, influenza epidemics are viewed as regularly occurring disease events, with temperate regions of the world reporting increases in influenza incidence during winter months [10,11]. Several studies based in North America and Europe incorporate annual seasonality of influenza as a known component of influenza dynamics [12–14]. This is a well-regarded pattern outside of influenza as well; other respiratory diseases with identified annual or near-annual patterns in temperate regions include RSV [15], common cold causing coronaviruses [16,17], and rhinoviruses [18].

For any pathogen, if periodic increases in incidence occur and align with the calendar year, a natural conclusion to draw is that the disease dynamics are influenced by annually recurring external factors. When periodic signals are observed in disease transmission, as is seen with influenza in temperate regions of the world, it is common to reinforce the notion of seasonality by using periodic events and phenomena to describe the dynamics of the disease, commonly through statistical or mathematical models. Within influenza or other respiratory viruses, this has been seen by using annual cyclic predictors such as climate patterns [7,19], school terms [20], and holidays [21] to predict disease incidence. In regions that experience cyclic patterns in weather, seasonal changes in human behavior typically follow. Seasonal behavioral changes can include some that affect influenza transmission, such as gathering in indoor spaces more frequently. Additionally, virus transmission in different climate settings, such as under different humidity and temperature conditions, have been examined [22,23]. While both of these potential mechanisms of transmission, social contact patterns and virus survival in the environment, have been studied previously, there currently is no consensus of which mechanism or set of mechanisms has the strongest influence on seasonal influenza patterns [24]. There is continuing debate whether observed seasonality relates more strongly to seasonal changes in weather patterns or to seasonal changes in human behavior [25,26].

In contrast, tropical regions of the world do not experience pronounced changes in temperature patterns and also do not appear to experience strong influenza seasonality, though a consensus has yet to be reached on this topic. In much of the tropics, influenza is seen throughout the course of the year rather than during particular parts of the year [1,7,27–29]. Less consistent evidence of influenza seasonality has been observed in tropical countries, including much

of Latin America [30], sub-Saharan Africa [31], and south and southeast Asia [32–34]. Some work has been done seeking to determine whether influenza follows annual or nonannual cyclic patterns; results of these studies have provided some evidence for annual cycles or nonannual cycles [35–38], while others have not found substantial evidence for cyclic patterns in influenza [39–41].

Despite less consistent seasonality in the tropics, many studies have aimed to relate influenza patterns to climate in the tropics. Some evidence has shown that environmental factors may influence influenza in both a tropical and temperate setting, but in different ways [28,42]. Differences in seasonal changes in precipitation and humidity as well as seasonal changes in behaviors such as social mixing may relate to the differences in distinctness of influenza seasons. Despite ample literature showing statistical associations between influenza (or other respiratory viruses) and seasonal covariates, notably temperature, precipitation, humidity, and wind speed [35,43–47], there is a dearth of research seeking to assess quantitative support for the existence of seasonal or annual patterns in the tropics [48].

In describing seasonality of influenza and, as a result, influenza seasons, definitions of seasonality can differ across studies. While many studies refer to seasonality as regularly occurring events during a particular time of year, others view seasonality as an annually recurring event, even if the time of year is not consistent. An important difference is that inconsistent seasonality, such as observing an outbreak every year, but at a different time each year, does not describe a system that is predictable through repetition. Here, we define "repeatability" as seasonality that occurs at a consistent time of year. In a repeatable system, knowledge of one season occurring is highly informative of when the next season will occur. Repeatability of influenza refers to having consistent and predictable seasonality at the same time each year.

This study aimed to determine the presence or absence of annual seasonality, or repeatability, of influenza in the tropics, using ten years of standardized surveillance data from Vietnam as a case study. Our starting hypothesis is that influenza is seasonal in Vietnam. We developed a mathematical model that describes influenza dynamics in Vietnam and incorporates parameters representing periods of increased transmission. Our statistical framework includes parameters representing peak timings of transmission and the repeatability of these peak timings is evaluated through Bayesian inference. Fitting the model to weekly influenza time series from four temperate locations (Netherlands, Denmark, and two regions of the United States) as well as northern, central, and southern Vietnam allows us to assess whether periods of increased transmission occur with greater or lesser repeatability and compare between temperate and tropical locations. The results of this study provide insight into whether influenza dynamics in Vietnam exhibit any annual or nonannual repeatable cycles.

## 2. Methods and materials

### 2.1. Influenza case data

In 2006, Vietnam's National Institute of Hygiene and Epidemiology (NIHE) established a sentinel surveillance system for influenza, consisting of 16 hospitals throughout northern, central, and southern Vietnam, 15 of which provided sufficient data for this study (Fig 1). This syndromic surveillance system collected weekly counts of patients with influenza-like illness (ILI), the number of PCR-confirmed influenza cases, and the number of PCR tests administered each week [49,50]. Among PCR-confirmed cases, the numbers of patients positive for subtype A/H1 (differentiating between 1977 lineage and pandemic 2009 lineage), subtype A/H3, and type B were included. Data were reported between 2006 and 2015, providing ten years of observation. Seven hospitals are located in northern Vietnam, including four in the city of Hanoi; four

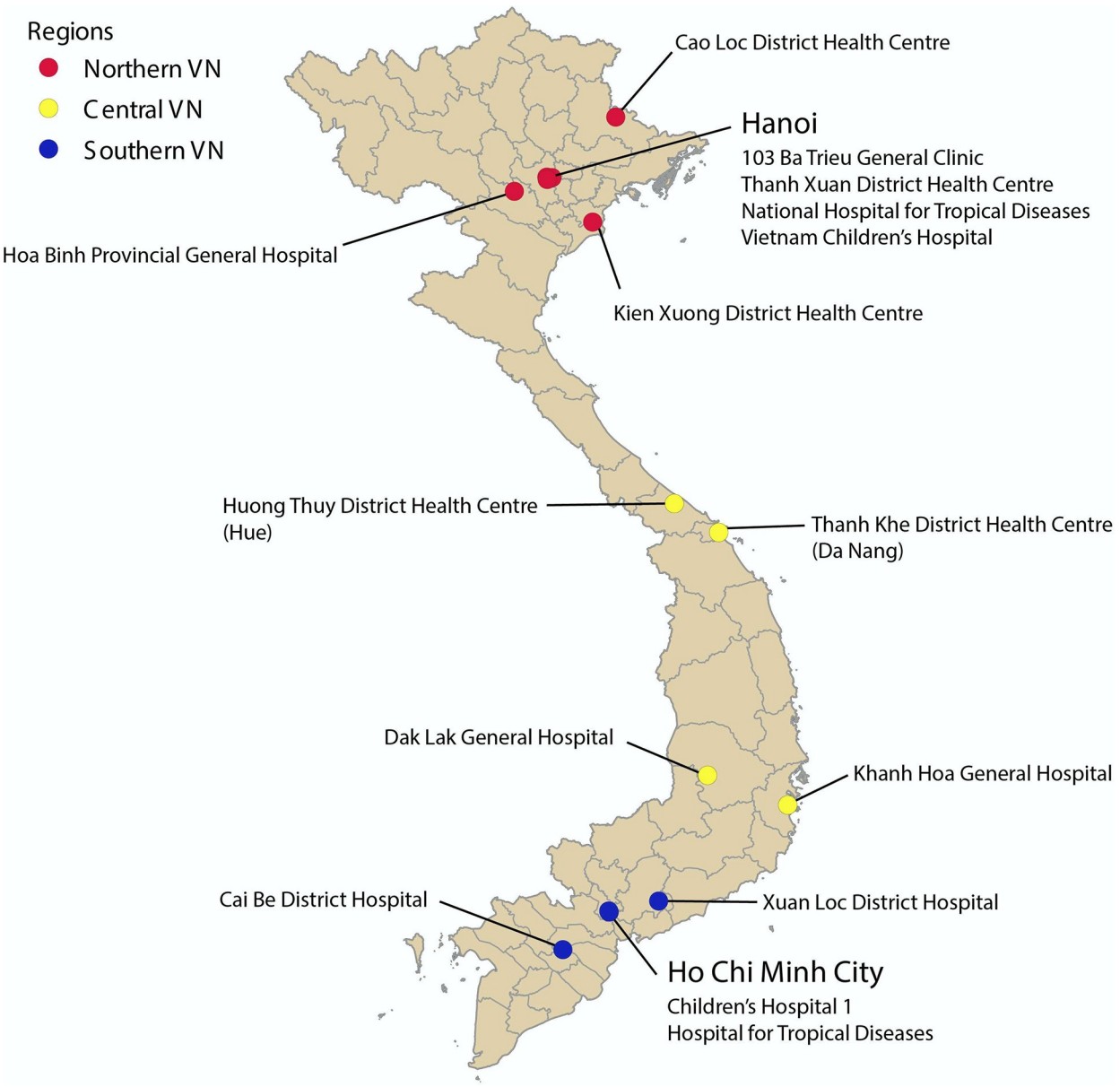

**Fig 1. Locations of sentinel hospitals in Vietnam.** Sites are colored by region: northern, central, and southern Vietnam. Sites located in the cities of Hanoi and Ho Chi Minh City are listed underneath the city name. License for map used can be found at gadm.org/license.html.

are located in central Vietnam; and four are located in southern Vietnam, including two in Ho Chi Minh City (Fig 1).

Reporting of ILI cases and testing data were inconsistent across hospitals throughout the period of observation. Nonuniform gaps in reporting as well as long-term changes in clinic visiting patterns over the ten-year period resulted in the appearance of artificial trends that relate more to administrative decisions to include or exclude certain hospital wards from reporting than due to actual incidence trends. For the purposes of data analysis, we detrended ILI cases by dividing each weekly case count by the one-year weekly average of cases centered at that week. This can be viewed as a log-scale transformation of a z-score, with $\zeta = e^z$; this $\zeta$-score approach preserves the relative sizes of case counts in neighboring weeks even after

detrending, allowing for mechanistic fitting of a traditional epidemic incidence curve. This approach of detrending using a one-year moving average preserves any potential annual cycles present in the data while acting as a high-pass filter for trends greater than one year. Though this detrending method (like z-scores) produces a unitless time series, it preserves short-term relative trends and is well-suited for incidence fitting. Preserving trends in consecutive weeks is most important for identifying trends and patterns in influenza dynamics. We then multiplied the detrended ILI by the percentage of tests that were positive for each of the three subtypes to obtain a detrended positive ILI (ILI+) for each of three subtypes [51]. We used this method of detrending for individual (sub)types of influenza as well as combined ILI.

Detrended ILI+ values for each hospital were then summed together within northern, central, and southern Vietnam, with each hospital weighted by the population of the province in which the hospital is located. One hospital in the southern Vietnamese city of Cai Be is located on the border of two provinces; therefore, its population weight was the average of the two province populations. Summing hospital totals minimizes the number of weeks with missing data. During weeks where some, but not all, hospitals provided data, the weighted average of ILI+ among hospitals presenting data were used. Separate analyses were run using detrended ILI+ data for five different virus types: (i) subtype A/H1, (ii) subtype A/H3, (iii) type A (sum of A/H1 and A/H3), (iv) type B, and (v) combined ILI+ (sum of A/H1, A/H3, and B). These five fits were performed for each of northern, central, and southern Vietnam.

We collected influenza surveillance data from the European Centers for Disease Control, which contain weekly case counts for all positive influenza between 2010 and 2019, but without specifying (sub)type. These data include the number of positive-testing influenza cases seen in healthcare settings and then reported to the ECDC. We used the sum of cases from sentinel (selected primary care facilities) and nonsentinel (including hospitals, schools, other primary care facilities) sources [52]. Data from the Netherlands and Denmark were the most complete data sets and therefore used in this study. We also collected weekly ILI cases for the United States (US) from the Centers for Disease Control and Prevention, which spanned between 2006 and 2015. Data from the US are present in ILINet, which reports the percentage of outpatient visits to healthcare providers for ILI [53]. We used data from two regions defined by the US Department of Health and Human Services which are located on opposite coasts of the US in order to increase variety of temperate locations. Region 1 spans the northeastern US, and region 9 encompasses the western US [54]. We detrended data from the Netherlands, Denmark, and the US using the same $\zeta$-score approach we used for the ILI+ data from Vietnam.

## 2.2. Patterns of seasonality

We conducted two standard time-series analyses to assess preliminary evidence of seasonality. The first consisted of an autocorrelation function (ACF), which identifies lag periods for which the time series has high autocorrelation. Lag periods with high autocorrelation indicate a repeated pattern of that length throughout the data. The second consisted of a wavelet decomposition using a Morlet wavelet, which fits an oscillating curve with multiple inflection points, with varying periodicities, to the observed time series data. This analysis reveals not only cycle lengths, but where within the time series these cycles can be found. The results of this analysis show where along the time series this function can fit the data given different period lengths. A particular periodicity fitting consistently over the entire time span provides evidence of stationarity, which can be interpreted as evidence of cyclic patterns.

## 2.3. Measuring repeatability

We developed an ordinary differential equations model to identify and quantify repeatability of influenza. Using a mathematical model to measure repeatability allows us (*i*) to include the natural mass-action dynamics of respiratory virus transmission and (*ii*) to add peak periods into these natural dynamics that can later be characterized as repeatable or not. The model takes the form of a SIRS model, which includes a sinusoidal time-varying transmission parameter, $\beta(t)$, that incorporates timings of potential seasons [55,56]. We define the model by

$$\frac{dS}{dt} = -\beta(t)IS + \frac{\gamma}{4}R_4$$

$$\frac{dI}{dt} = \beta(t)IS - vI$$

$$\frac{dJ}{dt} = \beta(t)IS$$

$$\frac{dR_1}{dt} = vI - \frac{\gamma}{4}R_1$$

$$\frac{dR_i}{dt} = \frac{\gamma}{4}R_{i-1} - \frac{\gamma}{4}R_i, i \in \{2, 3, 4\}$$

$$\beta(t) = \begin{cases} \beta_0[1 + a \cos^+\left(\frac{2\pi(\varphi_k - t)}{2\delta}\right)], \varphi_k - \frac{\delta}{2} \leq t \leq \varphi_k + \frac{\delta}{2} \\ \beta_0, otherwise \end{cases}$$

$$\varphi_{k+1} = \varphi_k + \tau_k$$

where $\cos^+$ is the positive-valued cosine function, $\varphi_k$ represent timings of peak transmission, and $\tau_k$ represent the time (in days) between $\varphi_k$ and $\varphi_{k+1}$. The $\tau_k$ parameters are free and independent of each other to allow for completely irregular and non-annual epidemics. It follows that, while the $\varphi_k$ values depend on each other in that $\varphi_{k+1} > \varphi_k$ by definition, the distance between the two values is independent of either value. The number of $\varphi_k$ 'peak timing' parameters (typically between 8 and 14 in our 10-year data set) was chosen a priori for each location and subtype based on visual examination of incidence trends followed by comparing preliminary model fits with different numbers of $\varphi_k$ parameters. The three models with 13 or 14 $\varphi_k$ were those for all ILI+ in northern, central, and southern Vietnam. Half (6/12) of the models for (sub)types in Vietnam and all four temperate models used 10 $\varphi_k$ parameters. In the model, $\delta$ represents the duration, in days, of increased transmission (i.e. the duration of the increased transmission), $\gamma$ represents the mean rate of waning immunity (reciprocal of duration of immunity in days), and $a$ represents the relative increase in transmission, where $a\beta_0$ is the amplitude of the cosine function. The locations of the $\varphi_k$ parameters (if consistent and repeatable) would correspond to annual seasonal dynamics. Regularly spaced peaks (i.e. low variance of timings between peaks) indicate strong annual seasonality, and higher variance of timings between peaks indicates weak annual seasonality.

The model also incorporates multiple recovery classes in order to reduce the variance in recovery time. Having multiple, sequential recovery classes prevents immediate loss of immunity, allowing immune-waning to follow a more realistic $\Gamma$-distribution rather than an

exponential distribution [57]. Four recovery classes were chosen, where immunity wanes only after exiting the fourth class in the sequence. In our data, when examining individual subtype data, periods of zero ILI+ exist. In order to facilitate time periods where zero prevalence occurs, a model behavior was added outside of the differential equations where ILI+ is set to zero based on a parameter $z$ (fit from the data) at which point all infected individuals immediately recover. The value of $z$ is set to be small so that this only occurs when an epidemic is ending, serving the purpose of preventing a very small, but nonzero, population from existing in the infected class to potentially infect others. When ILI+ falls below $z$, ILI+ is set to zero in order to prevent future transmission and end the epidemic. To then avoid extinction in the model, an immigration parameter was added that spontaneously inserts an influenza case from the susceptible population every $n$ days (fit from the data). This represents influenza transmission occurring outside of the study area and then being introduced. A reporting parameter was included to account for the proportion of influenza cases that occur but are not reported. We assumed that the reporting fraction remained constant throughout the time of observation.

## 2.4. Model fitting

To fit the model to observed data, we ran the model for a ten-year period as burn-in followed by a second ten-year period for model fitting and comparison to observed data. In this context, the ten-year burn-in period represents a time when the model produces incidence projections but is likely to show major fluctuations and not reflect the disease dynamics well. Comparing the data to the second ten-year span prevents these fluctuations that occur early in the model projection from influencing the model fit. Cumulative incidence variables ($J$) defined in the model were used to construct the model's weekly incidence $\Delta J$. We multiplied modeled ILI + by an estimated reporting parameter, $\rho$, and then fitted the output to observed data through a normal likelihood with fixed variance. Selection of a normal likelihood with fixed variance is a sum of squared errors fit, directly comparing the observed and predicted values. We also estimated the variance of the $\tau_k$ by estimating parameters representing their mean $\mu_\tau$ and standard deviation $\sigma_\tau$ from a normal distribution and comparing all $\tau_k$ to this normal distribution in the likelihood fitting. This standard deviation is an important parameter for assessing strength of seasonality; low standard deviation of $\tau_k$ (and thus of $\varphi_k$) provides evidence of strong seasonality because it indicates that the $\varphi_k$ repeat at the same time every year.

We fitted the model to data for each of subtypes A/H1 and A/H3, combined type A, type B, and all ILI+ in northern, central, and southern Vietnam using Markov Chain Monte Carlo (MCMC) sampling with the Metropolis-Hastings algorithm to estimate parameters [58,59]. We also fitted the model to ILI+ data from the Netherlands and Denmark as well as ILI data from the US to demonstrate the ability of the model to detect seasonal patterns in temperate countries. Parameters estimated include the transmission parameter ($\beta_0$), mean duration of immunity ($\gamma^{-1}$), magnitude of increased transmission ($a$), duration of increased transmission ($\delta$), timings of epidemic periods ($\varphi_k$), incidence level when epidemics end ($z$), and frequency of case importation ($n$). We selected diffuse priors (S1 Table). We assumed a five-day duration of infection [60].

The model we developed is sensitive to initial parameter values and can produce different outcomes through small changes. For example, holding other parameters constant and modifying the transmission parameter by 10% can lead to very different trajectories in incidence. Therefore, we conducted wide parameter space searches followed by close examination of changes in individual parameters to inform starting values prior to running the MCMC chains. This allowed smaller variances to be selected for the proposal distributions to increase

the probability of a proposal being accepted (S1 Table). To balance exploring parameter space during MCMC and efficient sampling, variance from proposal distribution was modified after every 50,000 attempted iterations to shrink it if there were too few accepted samples or expand if there were too many accepted samples. If between 10 and 90 percent of the last 50,000 attempted iterations were accepted, their variance was used to update the proposal distribution [61]. Otherwise, the variance of the proposal distribution was increased or decreased by 10 percent. Doing so allowed the acceptance rate of the MCMC to reach approximately five percent after seeing a starting acceptance rate below one percent.

We ran the MCMC for four independent chains for each of the five models using data from northern, central, and southern Vietnam as well as the models using data from the Netherlands, Denmark, and the US. A minimum of 50,000 burn-in samples were run, and then convergence was evaluated using visual inspection of trace plots and the Gelman-Rubin statistic [62]. After convergence was reached, an additional 50,000 samples were collected to generate posterior distributions, thinning to include every fifth sample. Results are presented from the chain with the highest median posterior likelihood value, with results from other chains used as robustness validation. The mode of the posterior distribution for each parameter with 95% credible intervals (CrI) are presented. The MCMC inference was run using R version 4.0.3 [63], using the 'mvtnorm' package for drawing parameters from the proposal distribution [64].

## 3. Results

Between 2006 and 2015, 4,799,182 outpatients attended the fifteen hospitals participating in Vietnam's National Influenza Sentinel Surveillance (1,644,488 in northern VN, 908,057 in central VN, 2,246,637 in southern VN), 515,598 of whom were ILI patients (214,238 in northern VN, 92,155 in central VN, 209,205 in southern VN). A total of 43,878 molecular-diagnostic tests were administered (18,190 in northern VN, 12,631 in central VN, and 13,057 in southern VN), of which 9,204 (21.0%) were PCR-positive for influenza virus. In northern Vietnam, 3,599 (19.8%) of tests were flu-positive, with 13.1% positive for influenza A, 5.8% positive for A/H1, 7.3% positive for A/H3, and 6.7% positive for influenza B. In central Vietnam, 2,634 (20.9%) of tests were positive, with 13.5% positive for influenza A, 6.9% positive for A/H1, 6.6% positive for A/H3, and 7.3% positive for influenza B. In southern Vietnam, 2,971 (22.8%) of tests were positive, with 14.8% positive for influenza A, 6.9% positive for A/H1, 7.9% positive for A/H3, and 8.0% positive for influenza B. A total of 22 tests were positive for both influenza A and influenza B, and eight tests were positive for influenza A without specification of subtype. These data points were discarded. The detrended time series of ILI+ stratified by (sub)type are shown in Fig 2.

### 3.1. Descriptive indicators of seasonality

Autocorrelation functions showed inconsistent and weak evidence of annual or nonannual cycles in the influenza data from Vietnam for all (sub)types as well as for combined ILI+. This is shown for southern Vietnam in Fig 3 and for all locations in S1 Fig. In contrast, the four temperate locations showed strong positive correlations for lags at and near one year (Fig 3). Similarly, the Morlet wavelets showed inconsistency in detecting repeated cycles in the data from Vietnam. While some periodicity was detected, the duration of periods differed across locations and (sub)types, and no periodicity was detected that consistently spanned the entire time period. The wavelets for the temperate locations, however, showed consistently strong evidence of seasonality, with 52-week cycles providing a strong fit over the entire time span of all four locations (Fig 3). These two descriptive analyses provide preliminary evidence that

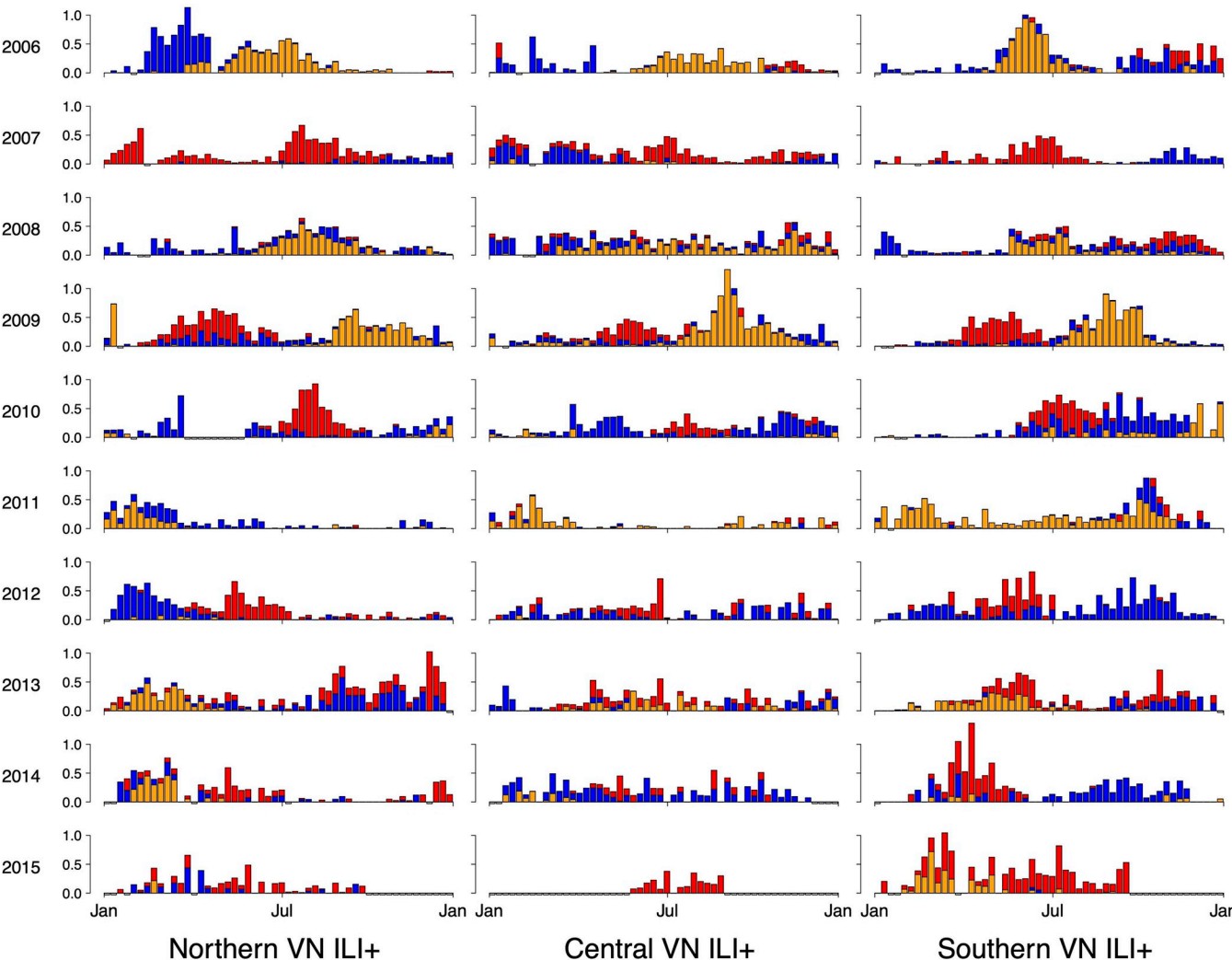

**Fig 2. Stacked bar chart of weekly time series of detrended ILI+.** Detrended ILI+ is shown for each year between 2006 (top row) and 2015 (bottom row) for each of northern (left column), central (middle column), and southern (right column) Vietnam. Colors denote (sub)types of influenza, with orange representing A/H1, red representing A/H3, and blue representing B.

seasonality of influenza is much weaker in Vietnam compared to temperate locations. Full results of the autocorrelation functions and wavelets are seen in S1 and S2 Figs, respectively.

### 3.2. Influenza peak seasonality and repeatability in Vietnam

The model fits showed close fits for the individual types and subtypes of influenza across the three regions of Vietnam for subtypes A/H1 and A/H3. The fits to combined ILI+, which used time series data with higher noise compared to the time series for individual (sub)types, were less accurate, and the fits for influenza B were generally less precise compared to the fits for A/H1 and A/H3 (Fig 4). This can be observed visually and by comparing observed data to model projected incidence. We collected mean absolute error values for all posterior parameter draws, standardized by the maximum observed data value to allow comparability between fits from different locations, and then compared the means of those values across models to compare which model fits were closest to the observed data. Values of standardized mean absolute

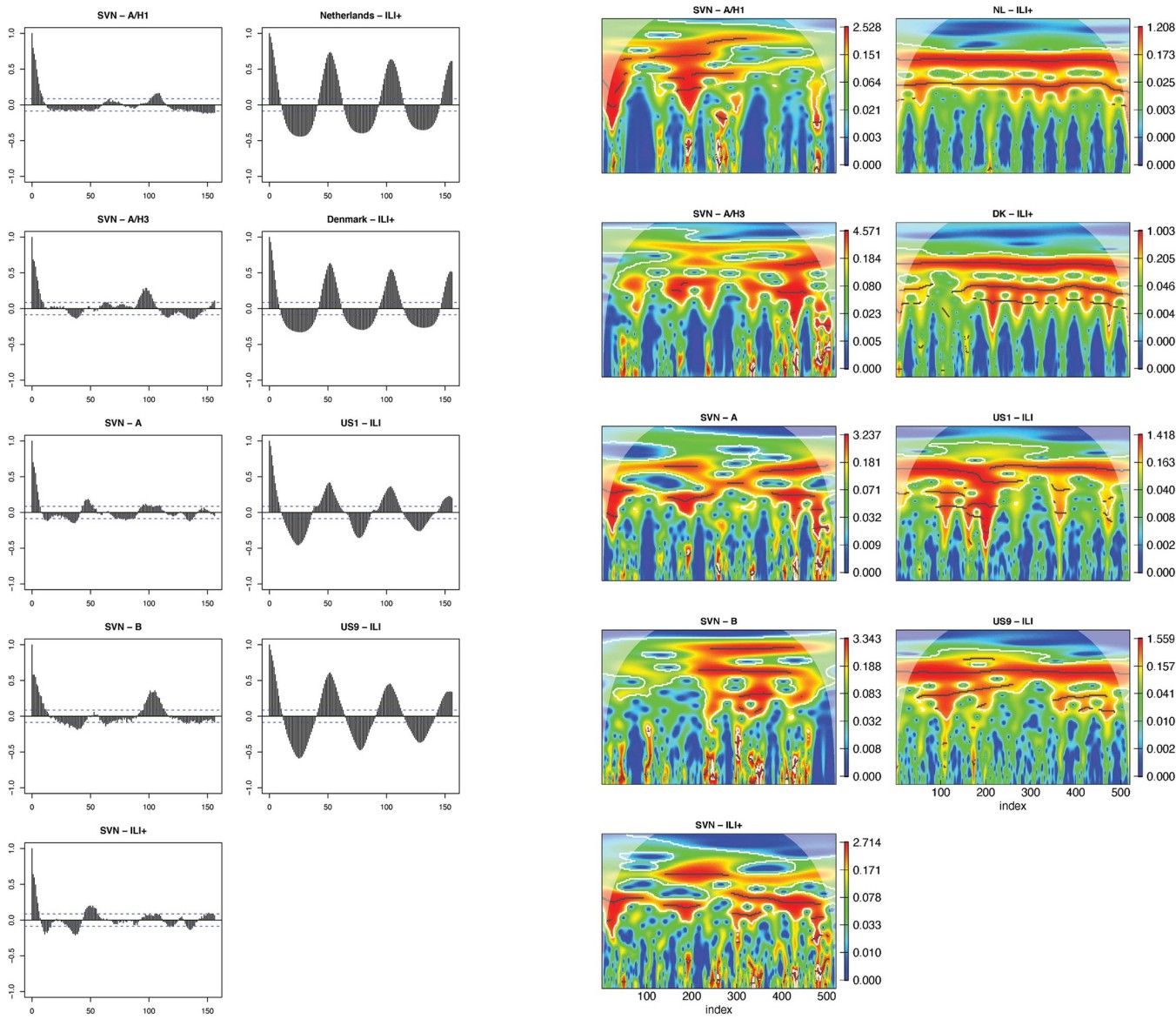

**Fig 3.** Results of autocorrelation functions (left side) and Morlet wavelets (right side) for influenza A/H1, A/H3, A, B, and combined ILI+ in Vietnam (left column) and for all ILI+ in four temperate locations (right column). Results for all locations and influenza (sub)types can be found in S1 and S2 Figs.

error (sMAE) are substantially lower for A/H1 and A/H3 (range of sMAE: 0.027–0.064) compared to the other three types (range of sMAE: 0.056–0.104) (S2 Table).

The estimates of the $\tau_k$ parameters, representing spacings between epidemic peaks, in Vietnam varied across locations and among subtypes. The average space between peaks ranged between 241 days (ILI+ in central VN) and 394 days (A/H1 in southern VN). The standard deviations of the time between peaks ranged between 64 days (B in southern VN) and 179 days (A/H1 in central VN) (Table 1, Fig 5). These high standard deviations in Vietnam indicate generally weak seasonality in transmission.

Converting the estimated $\tau_k$ parameters into $\varphi_k$ parameters (representing timings of peaks) shows whether certain times of year are most common for influenza peaks. These $\varphi_k$ parameters can be converted into calendar dates, and times of year when the $\varphi_k$ most frequently occur

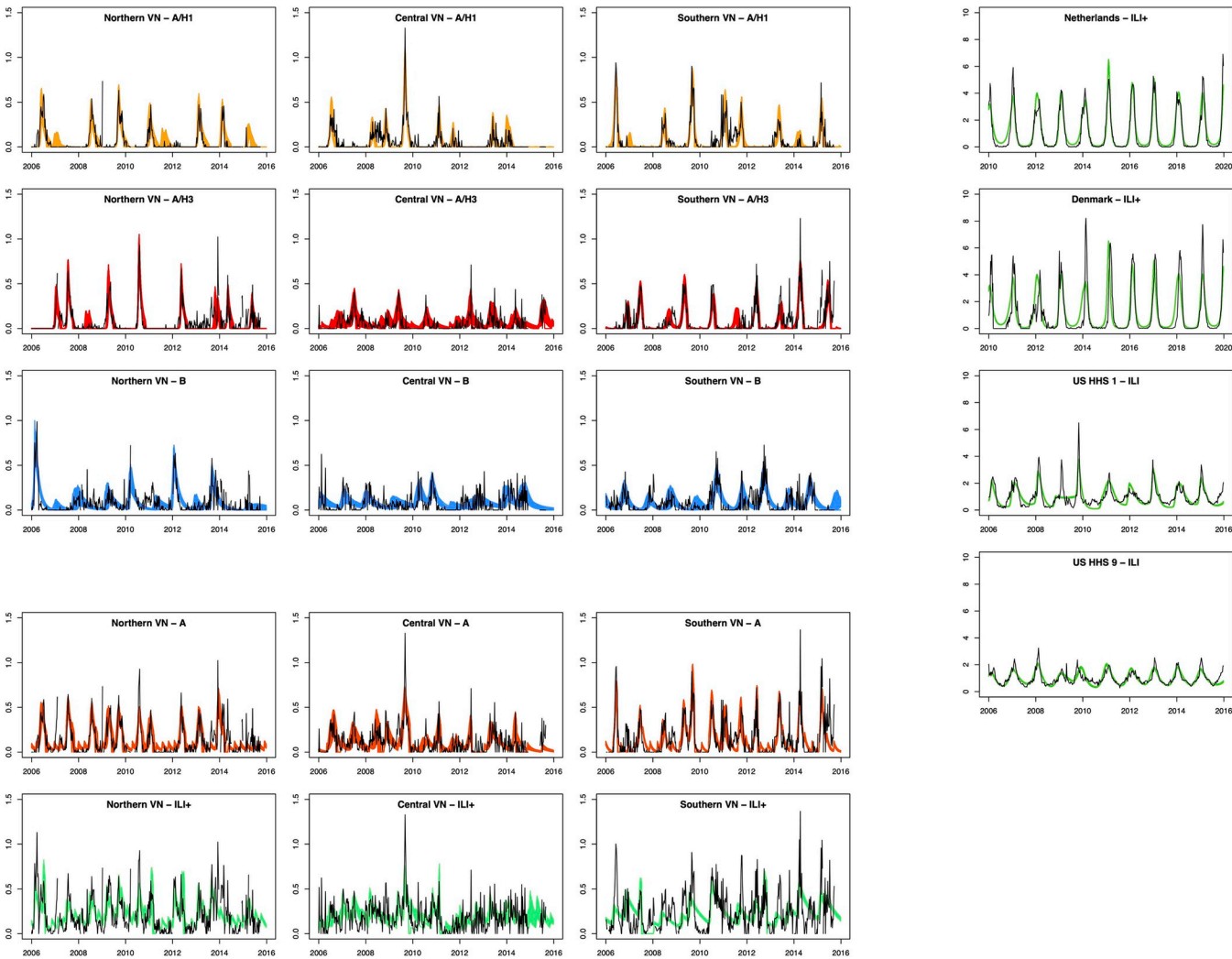

**Fig 4. Posterior model fits across influenza (sub)types as well as for combined ILI+.** Columns from left to right show model fits for northern Vietnam, central Vietnam, southern Vietnam, and four temperate locations. The colored bands show all fitted trajectories from the posterior parameter samples, with different (sub)types represented by different colors, and black lines show the observed data. The model fits for the better-fitting models for Vietnam, fitting data for A/H1, A/H3, and B, are vertically spaced from the worst-fitting models, which used data for combined (sub)types. The model fits for temperate locations are horizontally spaced from model fits for Vietnam.

would represent a regular "peak season" of influenza, if consistent. If influenza is repeatable, then the timings of peaks would be consistent, and the peak season would cover a short span of time. Across the three regions of Vietnam, peaks are less consistent, occurring throughout the course of the calendar year (Fig 6). Estimated peak seasons in Table 1 and Fig 6 represent the minimum time window that would encompass 90% of the posterior distribution of all $\varphi_k$. Within Vietnam, the shortest of these windows was 102 days, ranging between 22 Jun– 2 Oct for influenza B in southern Vietnam. The longest window was 302 days, ranging between 26 Oct– 24 Aug for ILI+ in central Vietnam. The median window was 244 days.

### 3.3. Influenza peak seasonality and repeatability in temperate countries

The model shows close fits to the data from the four temperate locations, both through visual closeness and low sMAE values described previously (range of sMAE: 0.036–0.061)

**Table 1. Estimated mean and standard deviation of spacings between transmission peaks with 95% credible intervals.** Estimated peak seasons represent the shortest time window that includes 90% of the posterior density of the $\varphi_k$.

| Location/subtype | Estimated mean of $\tau_k$ | 95% CrI | Estimated standard deviation | 95% CrI | Estimated 90% peak season |
|---|---|---|---|---|---|
| Northern Vietnam: A/H1 | 386 | [282, 506] | 124 | [88, 260] | 2 Dec– 11 Aug |
| Northern Vietnam: A/H3 | 376 | [249, 503] | 173 | [118, 270] | 28 Oct– 27 Jun |
| Northern Vietnam: A | 302 | [228, 398] | 106 | [73, 181] | 11 Nov– 14 Aug |
| Northern Vietnam: B | 374 | [272, 470] | 148 | [92, 193] | 29 Nov– 20 Aug |
| Northern Vietnam: ILI+ | 276 | [196, 365] | 126 | [90, 192] | 15 Jan– 19 Aug |
| Central Vietnam: A/H1 | 369 | [238, 554] | 179 | [129, 281] | 14 Dec– 29 Sep |
| Central Vietnam: A/H3 | 335 | [244, 468] | 143 | [99, 277] | 22 Oct– 7 Jul |
| Central Vietnam: A | 277 | [229, 335] | 70 | [51, 126] | 14 Mar– 5 Nov |
| Central Vietnam: B | 363 | [274, 400] | 68 | [50, 147] | 27 Jul– 11 Mar |
| Central Vietnam: ILI+ | 241 | [158, 331] | 124 | [93, 189] | 26 Oct– 24 Aug |
| Southern Vietnam: A/H1 | 394 | [276, 515] | 142 | [97, 252] | 3 Nov– 18 Aug |
| Southern Vietnam: A/H3 | 335 | [259, 418] | 98 | [65, 176] | 31 Jan– 28Jul |
| Southern Vietnam: A | 291 | [235, 340] | 76 | [56, 144] | 6 Feb– 27 Aug |
| Southern Vietnam: B | 366 | [319, 418] | 64 | [39, 127] | 22 Jun– 4 Oct |
| Southern Vietnam: ILI+ | 261 | [156, 303] | 115 | [74, 180] | 26 Jan– 27 Sep |
| Netherlands, ILI+ | 357 | [319, 398] | 57 | [43, 76] | 19 Nov– 27 Feb |
| Denmark, ILI+ | 378 | [356, 384] | 48 | [41, 62] | 7 Dec– 3 Apr |
| US Reg 1 ILI | 360 | [330, 394] | 46 | [30, 81] | 21 Sep– 14 Jan |
| US Reg 9 ILI | 358 | [333, 382] | 32 | [20, 60] | 12 Oct– 8 Dec |

(Fig 4, S2 Table). The time elapsing between transmission peaks was more consistent among temperate regions compared to Vietnam. The estimated average time elapsed was 357 days (95% CrI: [319, 398]) for the Netherlands, 378 days (95% CrI: [356, 384]) for Denmark, 360 days (95% CrI: [330, 394]) for the United States region 1, and 358 days (95% CrI: [333, 382]) for the United States region 9. The estimated standard deviations for the timing of peak transmission were 57 days (95% CrI: [43, 76]) for the Netherlands, 48 days (95% CrI: [41, 62]) for Denmark, 46 days (95% CrI: [30, 81]) in the United States region 1, and 32 (95% CrI: [20, 60])

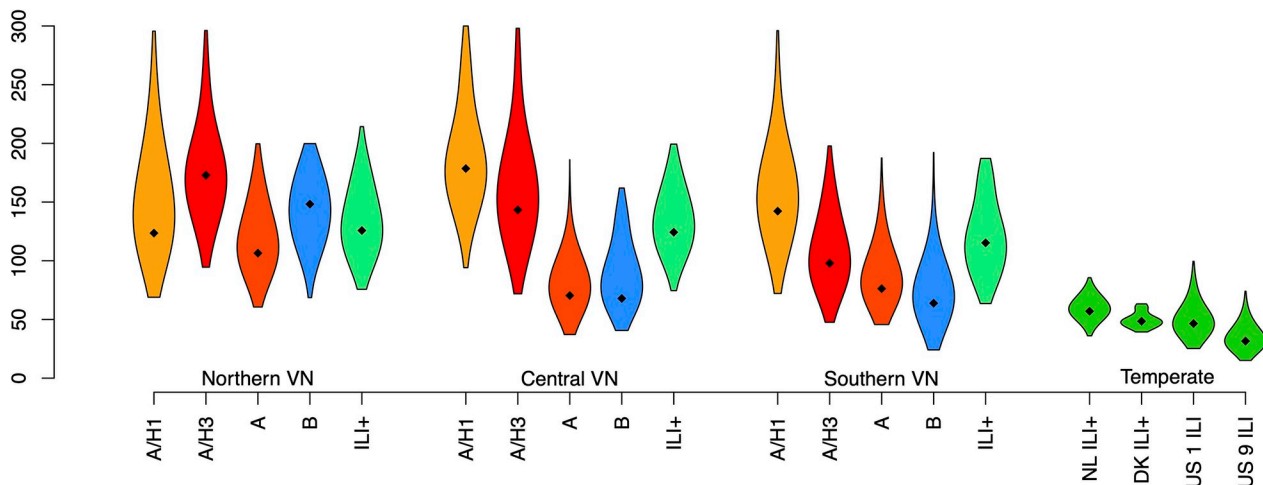

**Fig 5. Violin plots for estimated standard deviations for peak season timing ($\varphi_k$) (modes from posterior distributions, with 95% credible intervals) for A/H1, A/H3, combined A, B, and combined ILI+ in northern, central, and southern Vietnam.** These are compared to estimated standard deviations for timings of ILI+ peaks in the Netherlands and Denmark and ILI in two regions of the United States.

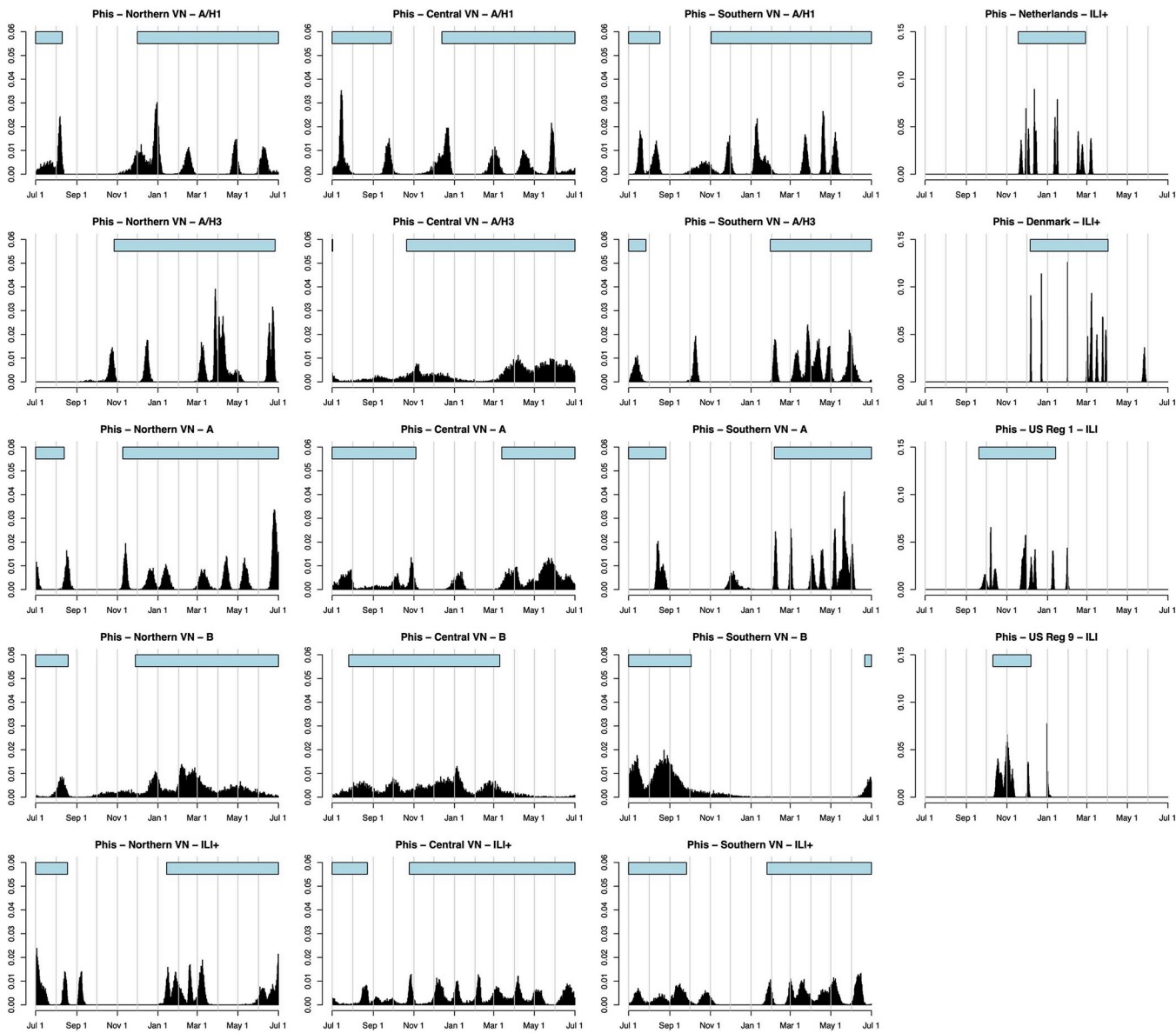

**Fig 6. Posterior densities for timing of peak influenza transmission, i.e. the posterior distributions of $\varphi_k$–the peak time–across all seasons included in the analysis.** Density is determined by frequency of $\varphi_k$ values in posterior MCMC samples corresponding to days of the calendar year, spanning between July 1 and June 30. Vertical lines show first days of calendar months. Horizontal bars above the distribution show the minimum time span that encompasses at least 90% of the posterior density, serving as a representation for a peak season.

in the United States region 9 (Table 1). Thus, the variation in peak timing in temperate regions (sd of about 35–60 days) is much smaller than the typical 100-day to 160-day variation in peak ILI+ seen in Vietnam.

As an alternate comparative approach, the posterior distribution of all peak timings can be constructed and compared between tropical and temperate areas, with peak season defined as the period encompassing 90% of the posterior density of φ, as described previously (Fig 6). In temperate zones, peak season is contained within a one-to-three-month time window whereas peak season in three regions of Vietnam would have to be defined as a 6–7 month period. Additionally, when separated out by subtype in southern and central Vietnam, the peak ILI

+ timings do not overlap, indicating that there is no preferred climatic period or school-term period that favors higher levels of influenza transmission. In temperate zones, all seasons encompassed late fall and winter months and lasted between 57 and 117 days, with the seasons for the Netherlands, Denmark, US region 1, and US region 9 spanning 19 Nov– 27 Feb; 7 Dec– 3 Apr; 21 Sep– 14 Jan; and 12 Oct– 8 Dec, respectively (Table 1, Fig 6).

Other parameters were shown to differ among the model fits comparing Vietnam to temperate settings. The fits from the Netherlands and Denmark showed notably higher baseline transmission parameter values, but lower relative transmission increases during outbreak periods. The estimated duration of epidemics was also estimated to be notably higher for A/H1 and A/H3 in Vietnam compared to the other model fits. Sporadic introduction of cases occurred more frequently among ILI+, A/H1, and A/H3 in Vietnam compared to the other model fits. The estimates for duration of immunity after infection were typically higher for infections of type A and its two subtypes compared to type B, particularly in northern and central Vietnam. Estimated duration of immunity was also typically lower for model fits for ILI + in Vietnam and the two European countries compared to the United States (S2 Table).

To examine the possibility of the 2009 H1N1 influenza pandemic affecting the results and conclusions of this study, we reran the models with the best fits from Vietnam (A/H1 and A/ H3) as well as the temperate models that encompass time from the pandemic (US1 and US9). We omitted data from the pandemic using three definitions: (i) June 2009 –August 2010, based on the start and end dates declared by the World Health Organization [65], (ii) July 2009 –December 2010, based on the start of the pandemic's effects on Vietnam (high community spread) [60] and following 18 months, and (iii) July 2009 –June 2012, based on the start of the pandemic's effects on Vietnam and following 36 months. While the model fits allowed epidemics to occur at various times throughout the period of omitted data (S4 Fig), the relative comparisons in variation in peak epidemic timings remained consistent (S5 Fig).

## 4. Discussion

This study aimed to address seasonality of influenza in Vietnam by determining its existence through a mathematical model able to test the hypothesis that seasonal repeatability does exist. Repeatability was defined here as the consistent and predictable generation of epidemics at the same time each year. Fitting the model to positive influenza incidence in temperate settings demonstrated the model's ability to detect seasonality when it exists through parameters representing the timings of periods with increased transmission. Applying this model to positive influenza incidence in Vietnam, using combined incidence as well as for individual (sub)types, showed much weaker evidence of repeatable seasonality through more widely varying timings of increased transmission. While the four temperate locations examined in this study showed that peak transmission occurs consistently during winter months, with timings between peaks having low variance, all three regions of Vietnam show peak transmission occurring throughout the year, typically with high variance in the inferred timings of a true "peak" if it were to exist.

Our finding of non-repeatable influenza in Vietnam is consistent with some research investigating seasonality of tropical influenza. Previous studies have also noted a lack of seasonal trends in tropical locations, whether by acknowledging that patterns were not identified [66– 68] or by describing seasons of influenza that do not show consistency in timing [37,69]. Nevertheless, other studies have found evidence of regular annual [19,41,70,71] and nonannual [28,72] cycles in tropical influenza and have recommended preparedness efforts such as vaccination based on these. Similar to the results of this study, other studies identifying seasonal patterns have noted that the strength of seasonality in tropical locations is not as strong as that

seen in temperate locations. Common seasonal patterns included yearly peaks occurring at different times of year [70] and observing two peaks per year [37,48,73]. Here, we show that while influenza epidemics may occur each year in Vietnam, they are not consistent or repeatable. By quantifying the comparatively large variance in timings of epidemic peaks in Vietnam (Fig 5), we show that knowledge of the timing of one epidemic peak does little to inform the timing of the following epidemic peak. Though it is not possible to choose a non-arbitrary threshold for repeatability in the estimated standard deviation, the trend of model fits from Vietnam showing higher standard deviations provides evidence against strong, repeatable seasonality.

We used a mathematical model to measure repeatability to evaluate seasonality through peak timings, with wavelets and autocorrelation functions used for preliminary evidence. The use of wavelets has been common for addressing influenza seasonality in previous studies [19,41] but the use of mathematical models has not. Other works directly addressing seasonality in the tropics have used various types of statistical models, including linear and logistic regression models [37,66,72], general estimating equations [69], or clustering methods [67]. Within these, it was common to incorporate sinusoidal functions to represent seasonal patterns [73], as we did with our time-varying transmission parameter. These non-mechanistic approaches provide evidence for or against annual patterns in time series, but they do not explicitly include the mass-action dynamics, immune acquisition, and immune loss that is known to occur in influenza epidemics. By directly assessing whether seasonality exists in a human-to-human pathogen transmission system, we avoided the inherent assumption made in previous analyses that the associations found between seasonal factors such as climate and influenza risk are consistent across temperate and tropical locations. Our study also benefitted from the use of a model that estimated both peak timing and the variability of peak timings. This allowed us to use a unified inference system for temperate and tropical influenza, to validate that consistent winter peak timing was robustly inferred in all temperate data, and to compute the variability in timing of a hypothesized influenza season in the tropics.

Assessments of seasonality are critical for tropical countries as they determine whether vaccine rollout can or should be timed to occur before an influenza season. The knowledge that regular cycles are weak or nonexistent is important for such preparedness efforts. In contrast to temperate locations, where vaccination and health messaging can effectively be targeted during autumn, there is not a clear time of year when vaccines can be prioritized in Vietnam for maximum protection. Because influenza can be seen year-round in Vietnam and other nearby tropical locations, identifying a single time of year to prioritize vaccination is less likely to be most effective, despite previous attempts based on data descriptions [39]. Similarly, knowledge of the timing of a current peak is not informative for predicting the timing of the following peak. Currently, despite an existing vaccine policy aiming to target some at-risk demographics [74], influenza vaccination is not common in Vietnam even among some high-risk demographics, particularly in rural areas [75,76], and the results of this study highlight a major challenge in implementing a vaccination strategy.

## 4.1. Differences in transmission intensity between temperate and tropical regions

A key comparison missing in the literature is whether the inherent community-level transmission of influenza virus (that is, one level beyond what is measured in household studies) is higher in tropical regions or temperate regions. This question would need to be answered separately for peak transmission and baseline transmission periods. Measures of population density, crowding [77], or mobility [78] are likely the most appropriate proxies for community

transmission levels that are typically observed for any respiratory virus. Our inferred dynamics are consistent with higher crowding/aggregation/transmission levels in temperate areas, and with temperate areas experiencing smaller changes in transmission amplitude. Our current study, however, is unable to identify this particular difference in inherent community-level transmission. Further work effectively comparing population mixing across the locations investigated in this study requires substantial resources and poses challenges in implementation.

The fits from the four temperate locations showed transmission parameters that were higher than those estimated across (sub)types in Vietnam. The temperate transmission parameters, along with the assumed five-day duration of infection, are consistent with a reproduction number between two and three (S2 Table). Previous estimates for the reproduction number in Europe and North America lie between one and two [79,80], though some estimates above two have been seen [81,82]. The fits for Vietnam, however, suggested basic reproduction numbers below, but near, one, increasing above one during the fitted periods of increased infectivity. Some evidence has shown that reproduction numbers globally trend higher in countries farther from the equator, and some tropical locations in Brazil have seen reproduction numbers below one [83–85]. An effective reproduction number that is consistently close to one in Vietnam would allow infections to persist in the population in lower quantities. Relative increases in transmission during epidemic periods were notably higher in Vietnam compared to temperate countries. While the United States, Denmark, and the Netherlands observed transmission increases of approximately ten percent, some model fits from Vietnam showed increases of over 50 percent. Over such a short time, true mechanistic reasons for an increase in transmission of nearly 50 percent are unlikely; these large relative increases in transmission, as well as the timings of them denoted by the $\varphi_k$ parameters, may instead represent irregular, less predictable events in Vietnam that can trigger an increase in transmission (e.g. the arrival of the virus to a more susceptible population pocket).

Many of the model fits for Vietnam show trajectory shapes that are less symmetric than a typical epidemic curve (compare to epidemic curves see for the four temperate locations) particularly when fitting ILI+ (Fig 4). This behavior was examined and was found to result from the combination of low estimates for the baseline transmission parameter along with high amplitude of forcing. After epidemic periods begin, $R_{eff}$ rises above one abruptly (this is an artefact of the model construction), allowing epidemic behavior to occur. After this period ends, $R_{eff}$ falls below one, but there are both adequate proportions of the population infectious and susceptible to sustain a slow decline in infections rather than a quick decline. A higher baseline transmission parameter would lead to higher peaks in transmission, and therefore a steeper decline in transmission when epidemic periods end.

### 4.2. Importance of methodological approach

While most fits across (sub)types showed high variance in time between epidemic peaks in Vietnam, fits for types A and B in central and southern Vietnam showed lower variances, more similar to those inferred from temperate locations (Table 1, Fig 5). These models, however, had substantially poorer fits (S2 Table). The model showed close fits to the observed data from Vietnam for the subtypes of influenza A as well as from the temperate locations, with the temperate locations showing closest and most precise fits (Fig 4) since different (sub)types synchronize into single seasons. One notable consideration is that the model used in this study simulates transmission of a single pathogen. This shows a limitation of the model in fitting data that are noisy rather than following typical epidemic trajectories. This study relies on the assumption that the models applied to individual (sub)types of influenza in Vietnam function

independently of each other since the model does not account for coinfection and cross-immunity among the three subtypes. This is a future extension of this work; a challenge in incorporating this was that it rarely allowed the subtypes to trade off while maintaining a single set of $\varphi_k$ parameters. The aim of this study focused on identifying annual or nonannual cyclic trends, for which the $\varphi_k$ parameters are critical.

Across models for different (sub)types and locations, estimated parameters differed in value, creating difficulty in inferring mechanisms of influenza. Notably, duration of immunity following infection was lower for influenza type B in Vietnam compared to type A or either of its subtypes, though there was low precision for most of these fits. Similarly, duration of immunity for ILI+ was typically lower, and there was a substantial difference in duration between the European locations and the US locations (S2 Table). These fits loosely align with previous estimates of immune duration of influenza [86]. It is worth noting that this parameter in practice can represent more than the immune system's retention of immune memory. Antigenic drift can occur at different rates for different (sub)types over time, evidenced by year-to-year differences in influenza vaccine effectiveness [87]. More importantly, inference of this parameter is affected by the implied attack rate in the model. Infrequent occurrence of a virus, as seen in our fits for subtype A/H1 in particular, can lead to an overestimation of immune duration; the length of time between infections may not be mechanistically caused by a long duration of immunity, but a long duration of immunity could fit the observed data well. This is further complicated by the fact that we fitted the model to unitless, detrended data, without the total population infected. Little has been done to directly study immune durations from influenza infections, as serological studies are needed for highest quality estimates.

Even within the (sub)types across the different regions of Vietnam, differences in model parameters were observed. For example, small differences were seen in transmission and duration of immunity of subtype A/H1 across the three regions of Vietnam (S2 Table). It is less likely that there are notable differences in virus characteristics across locations, but differences in transmission of the same (sub)type may result from different contact patterns or population densities. Similarly, epidemic-related parameters differed across locations such as timings of peak transmission, magnitude of transmission increase, and duration of transmission increase. These differences in epidemic characteristics are more likely to result from these differences in population densities and contacts across Vietnam and are indicative of the wide heterogeneity in influenza dynamics in Vietnam and lack of consistent seasonality; epidemic timing, amplitude, and duration are widely variable and unpredictable.

Differences seen in model fits between Vietnam and the temperate locations may also be affected by the influenza surveillance data sources. The influenza data from Vietnam were collected through a hospital-based sentinel surveillance system, the data from the Netherlands and Denmark were collected through both sentinel and passive surveillance, and the data from the US were collected through passive surveillance of hospital visits. This can lead to differences in data definitions and management. Further, the data from temperate locations are not available by individual (sub)types, preventing subtype-specific comparisons from being made. However, the data collected from the Netherlands, Denmark, and the US accurately represent the seasonal cyclic patterns known to occur, adequately showing that our model is capable of detecting seasonal trends when present.

## 5. Conclusions

Repeatable annual timing of influenza in Vietnam appears unlikely, as evidenced by an inferred high variance in peak time of epidemics. This is consistent with irregular, non-seasonal patterns observed previously in multiple tropical locations. Compared to temperate

locations, three regions of Vietnam showed notably higher variation in when increased influenza transmission occurs. These findings suggest that there are no strong annual or nonannual cycles in influenza transmission, and timings of increased transmission are not primarily driven by cyclic or predictable phenomena. It follows that management and prevention of influenza in the tropics presents a challenging direction for public health research as future epidemics will be irregularly timed and difficult to predict.

## Supporting information

**S1 Fig. Autocorrelation functions for ten years of weekly influenza incidence.** Columns from left to right show autocorrelation plots for northern Vietnam, central Vietnam, southern Vietnam, and four temperate locations.
(PDF)

**S2 Fig. Morlet wavelet decompositions for ten years of weekly influenza incidence data.** Columns from left to right show results for northern Vietnam, central Vietnam, southern Vietnam, and temperate locations.
(PDF)

**S3 Fig. Posterior distributions for all fitted parameters.** Individual model fits are shown with median values denoted by grey points and mode values denoted by red points. Locations and influenza (sub)types are indicated on the top left corner of each page. The location codes are: nvn–northern Vietnam, cvn–central Vietnam, svn–southern Vietnam, nl–Netherlands, dk–Denmark, us1 –United States region 1, us9 –United States region 9. The (sub)type codes are: h1 –subtype A/H1, h3 –subtype A/H3, a–type A, b–type B, c–combined ILI+.
(PDF)

**S4 Fig. Model fits when omitting the 2009 H1N1 pandemic.** Omitted times include the WHO definition of June 2009 –August 2010 (left column); 18 months from the start of the pandemic's influenza on Vietnam, July 2009 –December 2010 (middle column); and 36 months from the start of the pandemic's influence on Vietnam, July 2009 –June 2012 (right column).
(JPG)

**S5 Fig. Estimated standard deviations for the timings of epidemic peaks when omitting the 2009 H1N1 pandemic in model fitting.** The original fits are shown in the top panel. Omitted times include the WHO definition of June 2009 –August 2010 (second panel); 18 months from the start of the pandemic's influence on Vietnam, July 2009 –December 2010 (third panel); and 36 months from the start of the pandemic's influence on Vietnam, July 2009 –June 2012 (fourth panel).
(JPG)

**S1 Table. Prior distributions and standard deviations for proposal distributions for all parameters estimated through MCMC.** The row for tau_k indicates that all $\tau$ parameters were given the same prior distribution and standard deviation for the proposal distribution.
(PDF)

**S2 Table. Posterior model estimates with 95% credible intervals for all estimated parameters from all models.** Standardized mean absolute error (sMAE) values indicate relative goodness of fit across models.
(PDF)

## Acknowledgments

The authors thank the National Institute of Hygiene and Epidemiology in Vietnam for providing the data used in this study. The authors also thank the leadership and staff at the sentinel hospitals that collected the data used as well as Ephraim Hanks and Ottar Bjørnstad for providing methodological and substantive feedback on this work. The authors acknowledge the resources of Pennsylvania State University's Institute for Computational and Data Sciences Roar supercomputer for data analysis and management.

## Author Contributions

**Conceptualization:** Joseph L. Servadio, Maciej F. Boni.

**Formal analysis:** Joseph L. Servadio.

**Funding acquisition:** Joseph L. Servadio, Maciej F. Boni.

**Investigation:** Pham Quang Thai, Maciej F. Boni.

**Methodology:** Joseph L. Servadio, Marc Choisy, Maciej F. Boni.

**Software:** Joseph L. Servadio, Maciej F. Boni.

**Supervision:** Maciej F. Boni.

**Visualization:** Joseph L. Servadio, Marc Choisy, Maciej F. Boni.

**Writing – original draft:** Joseph L. Servadio.

**Writing – review & editing:** Joseph L. Servadio, Pham Quang Thai, Marc Choisy, Maciej F. Boni.

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
