## [Decision Letter · Decision Letter 0]

22 Feb 2023

Dear Dr. Servadio,

Thank you very much for submitting your manuscript "Repeatability and timing of tropical influenza epidemics" for consideration at PLOS Computational Biology.

As with all papers reviewed by the journal, your manuscript was reviewed by members of the editorial board and by several independent reviewers. In light of the reviews (below this email), we would like to invite the resubmission of a significantly-revised version that takes into account the reviewers' comments.

We cannot make any decision about publication until we have seen the revised manuscript and your response to the reviewers' comments. Your revised manuscript is also likely to be sent to reviewers for further evaluation.

Sincerely,

Claudio José Struchiner, M.D., Sc.D.

Academic Editor

PLOS Computational Biology

Virginia Pitzer

Section Editor

PLOS Computational Biology

Reviewer's Responses to Questions

**Comments to the Authors:**

Reviewer #1: uploaded as an attachment

Reviewer #2: Please see attached.

Reviewer #3: with pleasure I red your paper. I have a number of questions in particular related to the methodology:

- you compare sentinel hospital data from Vietnam with surveillance data from the Netherlands Denmark and the USA.

It is unclear where the data from the Netherlands Denmark and USA do come from? Hospital sector, primary care or community? The current presentation of data looks like a comparison between apples and pears.

- you include the pandemic season 2009-10 in your analysis: could you run a analysis without those years to see whether this has an effect on the results. It is known from literature that the 09 pandemic had impact on the next flu seasons in many countries.

Furthermore I am not convinced about the way you present your figure 3 > I do understand the results from Vietnam but the comparison with the ILI figures from NL DK and USA next to the Vietnamese figures confuses. What do the authors want to compare in this figure? Please clarify this also in the table text

The introduction and discussion are fine

**Have the authors made all data and (if applicable) computational code underlying the findings in their manuscript fully available?**

Reviewer #1: **No: **GitHub link provided doesn't exist

Reviewer #2: None

Reviewer #3: Yes

PLOS authors have the option to publish the peer review history of their article (what does this mean?). If published, this will include your full peer review and any attached files.

Reviewer #1: No

Reviewer #2: No

Reviewer #3: No
---

## [Decision Letter · Decision Letter 1]

29 Jun 2023

Dear Dr. Servadio,

We are pleased to inform you that your manuscript 'Repeatability and timing of tropical influenza epidemics' has been provisionally accepted for publication in PLOS Computational Biology.

Best regards,

Claudio José Struchiner, M.D., Sc.D.

Academic Editor

PLOS Computational Biology

Virginia Pitzer

Section Editor

PLOS Computational Biology

Reviewer's Responses to Questions

**Comments to the Authors:**

Reviewer #1: The authors have made revisions to the original manuscript to my satisfaction.

Reviewer #2: All comments were addressed satisfactorily.

Reviewer #3: The authors have succeeded in significantly improving the paper by better explaining and also expanding the methodology. There is now stronger argument for rejecting the hypothesis of repeatability for the Vietnamese flu seasons 2005-2016 despite still existing and for now unsolvable data inconsistencies and data limitations. The questions raised by the reviewers have been addressed in the right way. The paper can be accepted

**Have the authors made all data and (if applicable) computational code underlying the findings in their manuscript fully available?**

Reviewer #1: Yes

Reviewer #2: None

Reviewer #3: Yes

PLOS authors have the option to publish the peer review history of their article (what does this mean?). If published, this will include your full peer review and any attached files.

Reviewer #1: No

Reviewer #2: No

Reviewer #3: No

---

## [Editor Report · Acceptance letter]

13 Jul 2023

PCOMPBIOL-D-22-01686R1 

Repeatability and timing of tropical influenza epidemics

Dear Dr Servadio,

I am pleased to inform you that your manuscript has been formally accepted for publication in PLOS Computational Biology. Your manuscript is now with our production department and you will be notified of the publication date in due course.

With kind regards,

Zsofia Freund
